# The Benefits and Hazards of Intravitreal Mesenchymal Stem Cell (MSC) Based-Therapies in the Experimental Ischemic Optic Neuropathy

**DOI:** 10.3390/ijms22042117

**Published:** 2021-02-20

**Authors:** Yao-Tseng Wen, Yu-Chieh Ho, Yueh-Chang Lee, Dah-Ching Ding, Pei-Kang Liu, Rong-Kung Tsai

**Affiliations:** 1Institute of Eye Research, Hualien Tzu Chi Hospital, Buddhist Tzu Chi Medical Foundation, Hualien 970, Taiwan; ytw193@gmail.com (Y.-T.W.); jasonho60912@gmail.com (Y.-C.H.); josephyclee@mail.harvard.edu (Y.-C.L.); 2Department of Obstetrics and Gynecology, Hualien Tzu Chi Hospital, Buddhist Tzu Chi Medical Foundation, Hualien 970, Taiwan; dah1003@yahoo.com.tw; 3Institute of Medical Sciences, Tzu Chi University, Hualien 970, Taiwan; 4Department of Ophthalmology, Kaohsiung Medical University Hospital, Kaohsiung Medical University, Kaohsiung 807, Taiwan; 990288@gap.kmu.edu.tw; 5School of Medicine, College of Medicine, Kaohsiung Medical University, Kaohsiung 807, Taiwan; 6Institute of Biomedical Sciences, National Sun Yat-Sen University, Kaohsiung 804, Taiwan; 7Doctoral Degree Program in Translational Medicine, Tzu Chi University and Academia Sinica, Hualien 970, Taiwan

**Keywords:** mesenchymal stem cell, MSC-derived conditioned medium, HLA-DR expression, rodent model of anterior ischemic optic neuropathy

## Abstract

Mesenchymal stem cell (MSC) therapy has been investigated intensively for many years. However, there is a potential risk related to MSC applications in various cell niches. Methods: The safety of intravitreal MSC application and the efficacy of MSC-derived conditioned medium (MDCM) were evaluated in the normal eye and the diseased eye, respectively. For safety evaluation, the fundus morphology, visual function, retinal function, and histological changes of the retina were examined. For efficacy evaluation, the MDCM was intravitreally administrated in a rodent model of anterior ischemic optic neuropathy (rAION). The visual function, retinal ganglion cell (RGC) density, and neuroinflammation were evaluated at day 28 post-optic nerve (ON) infarct. Results: The fundus imaging showed that MSC transplantation induced retinal distortion and venous congestion. The visual function, retinal function, and RGC density were significantly decreased in MSC-treated eyes. MSC transplantation induced astrogliosis, microgliosis, and macrophage infiltration in the retina due to an increase in the HLA-DR-positive MSC proportion in vitreous. Treatment with the MDCM preserved the visual function and RGC density in rAION via inhibition of macrophage infiltration and RGC apoptosis. Conclusions: The vitreous induced the HLA-DR expression in the MSCs to cause retinal inflammation and retina injury. However, the MDCM provided the neuroprotective effects in rAION.

## 1. Introduction

In middle-aged people, the most common type of acute optic neuropathy is non-arteritic anterior ischemic optic neuropathy (NAION), with an incidence of 3.72 per 100,000 in Taiwan and at least 6000 new cases a year [1]. Transient nonperfusion or hypoperfusion of the optic nerve (ON) disc head (ONH) is the most common pathogenesis of NAION [2]. In addition, NAION is clinically recognized by visual loss and painless with optic disc swelling resulting in optic disc atrophy. Ischemic damage to ON causes a series of inflammatory responses and edema, ultimately leading to retinal ganglion cell (RGC) death and vision loss [2]. Effective treatments for NAION are yet to be established.

After the experimental induction of an ON infarct, blood–ON barrier disruption occurs within few hours, to be accompanied by the extrinsic macrophage infiltration and the resident microglia activation at the ischemic core of ON [3]. This inflammatory response also occurs in NAION. The activated macrophages/microglia provide the phagocytic ability to remove myelin debris. In addition, these activated macrophages/microglia release many different pro-inflammatory cytokines to induce severe inflammation, inhibit cell proliferation, and cause tissue damage [2,4]. These activated macrophages/microglia have been classified into M1-polarized type [5]. Macrophages/microglia can be switched from M1 to M2 phenotypes and can be classified by some specific markers [5]. M2 macrophages provide anti-inflammation, cell proliferation, and tissue repair capabilities [5]. Moreover, the activation of M2 microglia/macrophages has been reported to provide neuroprotective effects on some animal disease models [5,6,7]. Therefore, the inhibition of neuroinflammation may be vital for optic nerve protection in a rodent model of anterior ischemic optic neuropathy (rAION).

Mesenchymal stem cells (MSCs) are of medical interest because of their potential applications in tissue regeneration and immune modulation [8]. Recent studies reported that more than 2000 patients received autologous or allogeneic MSC treatments for different diseases [9]. In ocular diseases, MSC-based therapies provided a potential impact in alternative therapeutics, particularly for chronic diseases, such as retinal degeneration, optic nerve degeneration, glaucoma, and uveitis [10]. Besides, other therapeutic effects of MSCs have demonstrated their potential in the management of ocular surface disease and oculoplastics [10,11]. Some clinical trials using MSCs are in phase I/II have claimed the safety of intravitreal MSC administration in retinal and optic nerve diseases [10,11]. Preclinical studies have demonstrated that the neuroprotective effects of intravitreal MSC administration are due to anti-neuroinflammation and neurotrophic ability in some ocular disease models [12,13,14]. However, there is still a long way to go to demonstrating their efficacy and therapeutic effects in patients. In particular, lack of integration of MSCs and induction of reactive gliosis after intravitreal injection were also reported in some studies [11,15,16,17].

Although MSC therapy was considered relatively safe and efficient in many reports, the potential hazards of MSC-based therapies are being considered in current studies [9]. Some negative outcomes are not probably reported, but they may occur in certain conditions. The infusion of MSCs was demonstrated to possibly and dangerously inhibit antimicrobial immune response [18]. Although the anti-inflammation action of MSC treatment is addressed, the chance of infections may be increased by inhibition of immune response, particularly in patients receiving immunosuppressive therapy after allogeneic hematopoietic stem cell transplantation [19]. Notably, MSC transplanted into heart tissue may differentiate into other non-cardiac cells [20]. The unique niche of the heart contains several factors mediating stem cell fate and differentiation, which may result in a negative influence in clinical trials [20]. In addition, the size of MSCs in the monolayer culture might cause serious vascular obstructions after intravascular delivery and might not cross the blood–brain barrier [21]. Hypoxic and inflammatory reactions may lead to the differentiation of MSCs into osteoblasts and osteoblast-like cells [22]. If this process occurs in any soft tissue, it may directly contribute to heterotopic ossification formation. Thus, these potential risks have to be evaluated much more before clinical application. Some recent studies started to focus on MSC paracrine properties, including the release of various secretory factors and extracellular vesicles containing numerous regulatory compounds rather than MSC direct differentiation and cell replacement [23,24]. The induction of multiple repair processes in injured sites in vivo by secretory factors as well as the regulation of inflammatory response are considered to be the major effects of MSCs, which converts to a beneficial outcome of MSC-based therapies. Thus, treatment with MSC-derived soluble molecules not only provides beneficial effects but also prevents the potential risks.

Mechanical stress during the transplantation procedure or lack of an optimized dosage and protocol for transplantation may result in cell death at transplanted sites and potential risks. Recent clinical trials reported that suprachoroidal or sub-tenon MSC implantation are relative safe routes for retinal degeneration treatment [25,26,27]. In another clinical trial, the outcomes of subretinal MSC implantation were reported in 11 patients where there were no systemic complications, but 6 patients experienced ocular complications [28]. Up to now, epiretinal MSC transplantation has not been applied in clinic. Notably, the safety of intravitreal MSCs implantation is still controversial. A prospective, phase I clinical trial reported that intravitreal injection of autologous bone marrow MSCs into a patient’s eye does not meet safety standards. Major side effects of this therapy can include fibrosis and tractional retinal detachment (TRD) [29]. However, Aekkachai Tuekprakhon et al. reported that intravitreal injection of bone marrow-MSCs appears to be safe and potentially effective. All adverse events during the 12-month period required observation without any intervention. For the long-term follow-up, only one participant needed surgical treatment for a serious adverse event and the vision was restored [30]. Taken together, the proper MSC implantation routes in eye are considered be the sub-tenonian and suprachoroidal routes. The safety of subretinal, epiretinal, and intravitreal implantations of MSCs may need further evaluation before medical treatment to patients.

In this study, the safety of intravitreal human Wharton’s jelly mesenchymal stem cell (hWJMSC) application was investigated in normal Wistar rat by determining the fundus morphology, visual function, retinal function, and histological changes in the retina. In addition, the immunological response of hWJMSC transplantation was evaluated by staining HLA-DR, ED1, Iba1, and GFAP in the retina section. We found that allogeneic MSC transplantation in the vitreous chamber was able to trigger severe inflammation and retinal damage, which leaded visual function loss and retinal function loss. The efficacy of MDCM was evaluated in the rAION model. The intravitreal injection of MDCM provided promisingly neuroprotective effects via anti-inflammatory and anti-apoptotic actions.

## 2. Results

### 2.1. Intravitreal Delivery of hWJMSC Resulted in Retinal Venous Congestion and Breakdown of Retinal Architecture

Four weeks after intravitreal injection, we found that DiI-labeled hWJMSC aggregated at the central retina and caused severe venous congestion and retina distortion (Figure 1A). Besides, treatment with hWJMSCs dramatically disrupted the retina-layered structure in the transplanted eye (Figure 1B). The human mitochondria-labeled hWJMSCs were observed in the vitreous cavity with no sign of cell integration into the retina (Figure 1C).

### 2.2. Intravitreal Delivery of hWJMSCs Impaired the Visual Function and Retinal Function

The visual function was evaluated by using flash visual-evoked potentials (FVEPs) at the 4th week after intravitreal administration. The amplitude of the P1–N2 wavelet can indicate the RGC function in vivo. In this study, we determined the amplitude of the P1–N2 wavelet in the PBS-treated Wistar rat, which was 28.14 ± 13.68 μV. The amplitude of the P1–N2 waves in the hWJMSC-treated group was 13.67 ± 10.56 μV (Figure 2A). The amplitude of the P1–N2 in the PBS-treated group was higher than that in the hWJMSC-treated group (*p* = 0.0127).

The retinal function in the PBS-tread rat and in the hWJMSC-treated rat was evaluated by electroretinogram (ERG) at the 4th week after intravitreal administration. The scotopic ERG recordings demonstrated that a-wave and b-wave amplitudes were significantly decreased in the hWJMSC-treated group compared to those in the PBS-treated group (Figure 2B; a-wave, *p* = 0.0002; b-wave, *p* = 0.0062). In the photopic ERG analysis, a-wave and b-wave amplitudes were also significantly decreased in the hWJMSC-treated group compared to those in the PBS-treated group (Figure 2C; a-wave, *p* = 0.0002; b-wave, *p* = 0.0003).

### 2.3. Intravitreal Delivery of hWJMSCs Reduced the Number of RGC

After FVEP analysis, we found the transplanted hWJMSCs impaired visual function. Thus, we further evaluated whether transplanted hWJMSCs affect RGC viability. We quantified Brn3a-positive cells at the 4th week after intravitreal administration (Figure 3A). The densities of RGCs in the PBS-treated group and the hWJMSC-treated group were 38.86 ± 10.64 and 21.88 ± 6.57 cells/mm, respectively (Figure 3B). The RGC density was 1.81-fold lower in the hWJMSC-treated group than that in the PBS-treated group (*p* = 0.0007).

### 2.4. Intravitreal Delivery of hWJMSCs Induced Severe Retinal Inflammation

In the retina, retinal glial cells play an essential role in retinal immune response and maintain the homeostasis of the retina. We examined the changes of microglia and astrocyte cells in the retina by immunostaining of Iba1 and GFAP in the PBS-treated group and the hWJMSC-treated group. In the PBS-treated group, microglia are mostly located in the ganglion cell layer (GCL) and inner plexiform layer, and astrocytes presented in the GCL. In the hWJMSC-treated group, microglia are distributed throughout whole retina layers and migrated into hWJMSCs aggregates (Figure 4). In the hWJMSC-treated group, glia cells are distributed vertically throughout the entire retina (Figure 4). Infiltrating macrophages were associated with the activation of Müller glia cells during retinal inflammation. Hence, CD68, a phagocytotic macrophage marker, was used to investigate the macrophage infiltration. In the hWJMSC-treated group, the macrophage not only infiltrated into the vitreous cavity but also infiltrated into subretinal space compared to that in the PBS-treated group (Figure 4).

### 2.5. The Proportion of HLA-DR Positive hWJMSC Was Increased in Vitreous

In order to investigate the critical factor of retinal inflammation in the hWJMSC-transplanted eye, the HLA-DR-positive hWJMSC proportion was calculated by using Mander’s overlap coefficient and immunohistochemical staining of HLA-DR in the vitreous cavity. According to the calculation of Mander’s overlap coefficient, 50% of the DiI-labeled hWJMSCs colocalized with the HLA-DR positive cells. It indicated that the proportion of HLA-DR positive hWJMSC in the vitreous was elevated from 9.5% to 50% (Figure 5).

### 2.6. Treatment with hWJMSC-CM Preserved Visual Function

To evaluate the RGCs’ electrophysiologic function, FVEP was performed in the sham group, serum-free medium (SFM)-treated group, and CM-treated group. The P1–N2 amplitudes in the sham, SFM-treated, and CM-treated groups were 28.14 ± 13.68, 12.18 ± 5.36, and 24.04 ± 8.82 µV, respectively (Figure 6). The P1–N2 amplitude in the CM-treated group was 1.97-fold higher than that in the SFM-treated group (*p* = 0.0297).

### 2.7. Intravitreal Delivery of hWJMSC-CM Preserved RGC Density and Inhibited RGC Apoptosis

The Brn3a-positive cells were counted in the GCL to calculate the RGC density. The RGC densities in the sham, the SFM-treated, and CM-treated groups were 57.0 ± 9.1, 32.4 ± 8.0, and 48.4 ± 10.9 cells/mm, respectively (Figure 7).

The numbers of TUNEL-positive RGCs in the sham, SFM-treated, and CM-treated groups were 0.3 ± 1.04, 14.83 ± 7.67, and 0.33 ± 1.00 cells/mm, respectively. The apoptotic RGCs in the CM-treated group were significantly reduced by 44.9-fold compared to those in the PBS-treated group (Figure 7, *p* = 0.0001).

### 2.8. Intravitreal Delivery of hWJMSC-CM Inhibited the Neuroinflammation in the Optic Nerve

For evaluation of the neuroinflammation in the optic nerve, microglia activation and macrophage infiltration (ED1-positive cell) were investigated by staining with Iba1 and ED1 four weeks after ON infarct (Figure 8A). Treatment with CM reduced the stoma size of microglia (Iba1-positive cell) by 3.2-fold compared to treatment with SFM in rAION. In the sham group, there were 12.7 ± 6.9 Iba1-positive cells and 2.1 ± 1.3 ED1-positive cells per high-power field (HPF) in the ON section. In the SFM-treated group, 43.5 ± 13.4 Iba1-positive cells and 27.6 ± 8.3 ED1-positive cells per HPF was found in the injured area of the ON. In the CM-treated group, the number of Iba1-positive cell and ED1-positive cell per HPF was 22.1 ± 10.5 and 8.2 ± 4.3 in the ON section, respectively. The density of Iba1-positive cell was 1.97-fold lower in the CM-treated group than that in the PBS-treated group (Figure 8B; *p* = 0.029). The number of ED1-positive cells per HPF was 3.4-fold lower in the CM-treated group than that in the PBS-treated group (Figure 8B; *p* = 0.029).

## 3. Discussion

The present study demonstrated that intravitreal delivery of hWJMSCs is not safe to develop an alternative therapy for retina and optic nerve diseases because allogeneic MSC transplantation leads to visual function loss, retinal function loss, RGCs loss, retinal inflammation, retinal restructuration, and venous congestion. The main reason for these detrimental effects is that the transplanted hWJMSCs were induced to express HLA-DR in the vitreous cavity and trigger severe retinal inflammation. However, the treatment with hWJMSC-derived CM provided promising neuroprotection results in the experimental optic nerve ischemic model. Intravitreal injection of hWJMSC-derived CM preserved the visual function and RGC density. Besides, RGC apoptosis and optic nerve inflammation were significantly inhibited by applying CM treatment. Thus, we considered that allogeneic MSC-derived CM is a potential candidate to develop neuroprotective therapy for NAION patients.

Although lots of promising and beneficial findings were reported to support MSC therapies [10,11,12,13,14], some adverse effects were demonstrated in recent studies, including detrimental outcomes in stem cell therapy against eye diseases [15,16,17]. A previous report found that retinal vascular deterioration and severe vision impairment were induced after intravitreal injection of autologous BMSCs in patients with advanced retinitis pigmentosa [29]. Interestingly, one animal study proved that intravitreal injection of 2 × 10^4^ rat bone marrow-derived MSCs and or human adipose-derived stem cells leads to loss of pericytes and increased formation of acellular capillaries [30]. These findings indicated that allogeneic or autologous MSC transplantation in the vitreous cavity might cause damage to retinal vessels. In our findings of fundus photography, venous congestion was clearly observed in the hWJMSC-treated eyes. Venous congestion can lead to distention, edema, stasis, ischemia, and cellular death in the retina. This severe damage on the retinal vessel may result from the higher cell number of transplanted MSCs (1 × 10^5^ hWJMSCs) in the vitreous cavity. In contrast to those protective outcomes in early reports, we considered that intravitreal administration of MSCs is not safe for retinal vessels in both normal and diseased retinas. However, previous findings demonstrated that MSCs provided vasoprotective and proangiogenic effects in experimental animal models of diabetic retinopathy and other retinal diseases [31,32]. Taken together, we suggest that MSCs might have bidirectional effects, protective and destructive, on the retinal vessels in certain conditions.

Noteworthy, the retinal architecture was dramatically disrupted after hWJMSC transplantation in this study. However, we did not observe any transplanted WJMSCs migrate into the retina to change the retinal architecture. Similar findings were reported by Thomas et al., who demonstrated that the existence of the inner limiting membrane completely blocked the passage of transplanted MSCs into the retinas [16]. This indicated that the architectural changes in the retina may be induced indirectly by the hWJMSC-triggered severe immune response rather than direct damage from the transplanted hWJMSCs. We further evaluated the retinal inflammatory response in the hWJMSC-transplanted eyes. We found that there was severe microgliosis, astrogliosis, and macrophage infiltration in the retina. Many reactive microglial cells, astrocytes, macrophages existed in the retina tissue, which may lead to strong inflammatory impact and inflammation-induced damage in the retina. Jose E Millán-Rivero et al. also reported that intravitreal transplant of hWJMSCs into rat retinas induces a microglial reaction and retinal restructuration [11]. However, they claimed that intravitreal delivery of hWJMSC provided the neuroprotective effects in the axotomized rat retina due to their release of anti-inflammatory and neurotrophic factors [11]. Hongpeng Huang et al. reported that intravitreal administration of MSCs induced activation of retinal glial cells, and inflammatory response in normal and diseased retinas [31]. This indicated that the use of MSC therapy in the treatment of retina and optic nerve disorders is accompanied by the risk of severe ocular inflammation. We considered the controversial findings in the anti-inflammatory capability of hWJMSCs due to the higher cell number of transplanted hWJMSC in our study. More transplanted MSCs in the vitreous may trigger more severe inflammation in the retina. Besides, treatment with hWJMSC in the different animal models may generate diverse outcomes. Thus, the application of MSC therapy intravitreally must be approached with lots of caution in patients.

MSCs generally lack HLA-DR expression, while this is needed for antigen presentation to CD4 T-cells [33]. Our flow cytometric analysis revealed that 9.7% of the hWJMSCs expressed HLA-DR antigen (Appendix A). A previous report that activation of MSCs by IFN-γ increased the expression of HLA-DR without decreasing the expression of markers that characterize MSCs [34]. Therefore, intravitreal MSC administration in the diseased eye may induce more HLA-DR-positive MSCs than the normal eye and may increase the risk of enhancing inflammatory response. In addition, Marta Grau-Vorster reported that the expression of HLA-DR is dynamic in MSC cultures [35]. Therefore, they suggested that the use of HLA-DR to be a negative marker of MSC does not provide any additional value to quality control panels [35]. We also found that 50% of the transplanted hWJMSC expressed the HLA-DR antigen. It suggested that the vitreous cavity is not a suitable environment for the MSC niche. Taken together, we considered that the HLA-DR expression in MSCs is an environment-dependent process and is difficult to control this transformation in vivo. However, HLA-DR expression in the allogeneic MSCs may induce allogeneic rejection in the host. Especially, interspecies MSCs transplantation might cause much severe inflammatory response because the diseased ocular environment in these animal models might not be identical to the human environment. We considered that the autologous MSCs implantation in the vitreous cavity may prevent this severe inflammatory response. Thus, intravitreal injection of autologous MSCs in a safe and healthy manner needs further investigation.

The major beneficial outcomes of MSCs are considered to stimulate signaling pathways involved in proliferation and anti-inflammation by secretory factors, including the release of secretory factors and extracellular vesicles [23]. In this study, we found that the intravitreal injection of hWJMSC-derived CM effectively inhibited macrophage infiltration into the optic nerve and microglia activation at the optic nerve in the experimental model of optic nerve ischemia. In addition, we found treatment with the hWJMSC-derived CM preserved visual function and prevent RGC apoptosis after optic nerve infarct. Taken together, we considered that the hWJMSC-derived CM provided the neuroprotective effects after optic nerve infarct via anti-inflammatory action and anti-apoptotic action. Besides, we did not observe any change in retinal architecture after the intravitreal hWJMSC-derived CM administration. Thus, we believe that the use of hWJMSC-derived CM for retina and optic nerve disorders may provide promising and positive outcomes.

## 4. Materials and Methods

### 4.1. Animals

In this study, we used adult male Wistar rats weighing 150–180 g to evaluate the safety and the efficacy of MSC-based therapies. The rats were purchased from the breeding colony of BioLASCO Co., Taipei, Taiwan. Animal care and experimental procedures were performed in accordance with the Statement for the Use of Animals in Ophthalmic and Vision Research. All animal experiments were approved by the Institutional Animal Care and Use Committee (IACUC) of Buddhist Tzu Chi General Hospital. The IACUC approval number and date were 106-21-2 and May 28, 2019, respectively. 

### 4.2. Study Design

In the present study, we investigated the safety of intravitreal hWJMSC administration in the normal Wistar rat. The left eye of the rat received one intravitreal injection of hWJMSC suspension (5 μL, 2 × 10^4^ cells/μL, *n* = 12) or PBS (5 μL, *n* = 12). Four weeks after intravitreal injection, the hWJMSC-transplanted rat and the PBS-treated rat were analyzed by using fundus photography, FVEP, ERG, and IHC analysis to evaluate the fundus morphology, the visual function, the retinal function, and the histological changes in the retina.

To evaluate the efficacy of the hWJMSC-derived CM in rAION, 36 rats received rAION treatment and received an intravitreal administration of either hWJMSC-derived CM (5 μL; *n* = 12) or serum-free medium (5 μL; *n* = 12). Another 12 rats in the sham group received laser treatment without the use of photosensitizing agents. On the fourth week post-infarct, visual function was assessed by performing an FVEP analysis; the RGC density was measured by conducting the Brn3a-positive cell in the RGC layer. In situ TdT-dUTP nick end-labeling (TUNEL) Assays in the RGC layer and immunohistochemistry of ED1 (a biomarker of macrophage), Iba1 (a marker of microglia), GFAP (a marker of astrocyte) expression in the ON were also conducted (Figure 9). All the animals survived until the end of the treatment without any complication. The rats were euthanized using CO_2_ insufflation at the 4th week post-infarct.

### 4.3. hWJMSCs Culture and Collection of Conditioned Medium

HWJMSCs were isolated from a healthy human umbilical cord; the cells were kindly provided by Dr. Dah-Ching Ding in Hualien Tzu-chi medical center. The hWJMSCs used in this study were approved by the Research Ethics Committee of Buddhist Tzu Chi General Hospital, and written informed consent was obtained from all participants (Institutional Review Board 100–166). All experiments related to the hWJMSCs were followed by the institutional guidelines and approved by the Research Ethics Committee, Hualien Tzu Chi Hospital. The cells were cultured and expanded in low glucose Dulbecco’s modified Eagle’s medium supplement with 10% fetal bovine serum and 1% penicillin. Three days before transplantation, hWJMSCs were transferred and maintained in MSC serum-free medium (NutriStem^®^ XF Medium, Sartorius, Germany) for two passages. The culture was maintained at 37 °C and in 5% CO_2_. The cultured hWJMSCs were characterized by flow cytometry analysis (Figure 1). The conditioned medium (CM) was collected after 2 days of culture. The CM was further concentrated 30 times using a 3 kDa molecular weight cut-off centrifugal filter (Amicon-Ultra 15, Merck Millipore, Tullagreen, Cork, Ireland), then stored at −80 °C. Before the transplantation, hWJMSCs were incubated and labeled with fluorescent cellular membrane dye DiI (Molecular Probes, Carlsbad, CA, USA), then were resuspended in phosphate buffer saline (PBS) at the concentration of 2 × 10^4^ cells/μL.

### 4.4. Fundus Photography

The transplanted MSCs and the fundus morphology were monitored in the PBS-treated rats and hWJMSC-treated rats by using Micon IV fundus camera (Phoenix Research Labs, Pleasanton, CA, USA). Both bright-field and fluorescent filter were used to observe the DiI-labeled hWJMSCs on the retina at day 28 post-intravitreal administration.

### 4.5. Flash Visual-Evoked Potentials (FVEPs)

FVEPs were recorded in both eyes to examine the pattern of FVEPs. This procedure was approved by the Institutional Animal Care and Use Committee (IACUC) and was described in detail in our previous studies [36,37]. In brief, after administering the general anesthesia, the skin covering the skull was incised, and the brain surface was exposed using a dental drill. Two screw implants connected to active (positive) electrodes were fixed at the primary visual cortex region of both hemispheres. One screw implant, connected to a reference (negative) electrode, was fixed at the frontal cortex region. The ground electrode was placed in the rat’s tail. A visual electrodiagnostic system (UTAS-E3000; LKC Technologies, Gaithersburg, MD, USA) was used to measure the photopic FVEPs. The settings used were as follows: background illumination off, a flash intensity of 30 cd.s/m^2^, single flash with a flash rate of 1.02 Hz, and a test average of 64 sweeps. To evaluate the visual function, the latency of P1 wave, referred to as the first positive-going wavelet, and the amplitude of the P1–N2 wave in each group was compared.

### 4.6. Electroretinogram (ERG)

To evaluate the retinal functions, ERGs were performed four weeks after transplantation. Animals were placed in the dark for at least 12 h. A gold wire loop was placed on the cornea as the positive electrode, the negative electrode was placed under the animal scalp, and the ground electrode was inserted to the tail (Colordome Ganzfield, Diagnosys LLC, Lowell, MA, USA). The parameter of ERGs was provided by the manufacturer. In this study, both scotopic and photopic ERGs were recorded. To avoid noise signal interference, the contralateral eye was covered by a black cap. The a- and b-waves were measured to check the functions of retinal photoreceptor and bipolar cells, respectively.

### 4.7. Eyeball Preparation and Sections

Rats were euthanized by a gradual increase of CO_2_ concentration (5 L/min) inside the sealed cage. The enucleated eyeballs along with ON were removed from the surrounding connective tissues, muscles, and cornea by microscissors to improve the fixation process [36,37]. Six eyeballs per group were then transferred to 4% paraformaldehyde for 1 h at room temperature. Then, the eyeballs were cryoprotected in 30% sucrose and stored at 4 °C overnight until they settled at the bottom of the 15 mL tube. Both 4% paraformaldehyde and 30% sucrose volume were ten times greater than the volume of the eyeball. The eyeballs were then transferred into plastic molds and embedded in OCT compound. The molds were then allowed to completely freeze using liquid nitrogen. These frozen bricks were then transferred to cryostats at −20 °C, and 10 µm thick sections were performed. The sections were preserved at −20 or −80 °C for further analysis.

### 4.8. Immunohistochemistry of BRA3A-, ED-1-, Iba1-, GFAP- and HLA-DR-Positive Cell

The procedure of immunohistochemistry was described in detail in our previous reports [36,37]. In brief, six eyeballs in each group were detached and transferred to 4% paraformaldehyde to fix the protein components for 6 h. The fixed eyeballs were transferred into 30% sucrose and stored at 4 °C until they settled to the bottom of the tube as a result of dehydration. The OCT (Tissue-Tek, Sakura, Torrance, CA, USA) embedded eyeballs were sectioned longitudinally to 20 µm thick by using a cryostat (Leica Microsystems, Wetzlar, Germany). Three sections per eyeball were immersed in PBS to remove the OCT compound for one specific staining. The PBST with 5% BSA was used to block the cryosection at room temperature for 1 h. The primary antibodies, anti-Brn3a, anti-ED-1, anti-Iba1, anti-GFAP, and anti-HLA-DR (1:100; Abcam, San Francisco, USA) were applied in 4 °C overnight. Secondary antibodies were then incubated at room temperature for 1 h. DAPI staining (1:200; Sigma-Aldrich, St. Louis, MO, USA) was used to detect the number of nuclei in all sections. Imaging was acquired at whole retina or ON head by using a confocal microscope. We manually counted the Brn3a-positive cells in the RGC layer by randomly selecting 6 high-power fields (HPFs, 200×) and calculating the average to represent the number of TUNEL-positive cell in one section (*n* = 6 per group). The number of Iba1-positive cell was counted manually at ON head. For comparison, Iba1-positive cells were counted in six HPFs (magnification: 200×) at ON lesion sites.

### 4.9. In Situ TUNEL Assay

The protocol was provided by the manufacturer (Promega Corp, Madison, WI USA). Three sections per eyeball were immersed in PBS to remove the OCT compound. To increase permeability, the tissues were incubated with proteinase K at room temperature for 10 min. Samples were transferred to equilibration buffer and then incubated with recombinant terminal deoxynucleotidyl transferase (rTdT) reaction buffer at 37 °C for 1 h. To stop the rTdT reaction, the samples were incubated in SSC solution for 15 min at room temperature. Samples were then immersed with PBS and washed 3 times. Slides were then mounted with DAPI mounting medium and sealed the cover slide with DPX mounting medium. Imaging was acquired from central retina to mid-peripheral retina by using a confocal microscope. We manually counted the TUNEL-positive cells in the RGC layer by randomly selecting 6 high-power fields (200×) and calculating the average to represent the number of TUNEL-positive cell in one section.

### 4.10. rAION Induction

The detailed procedure of rAION induction was described in our previous report [36,37]. In brief, Rose bengal (RB) (Sigma-Aldrich, St. Louis, MO, USA) was administered intravenously through the tail vein (2.5 mM RB in PBS and 1 mL per animal weight) after general anesthesia. The rats in the sham group received an argon green laser treatment (MC-500 multicolor laser, Nidek Co., Ltd., Tokyo, Japan) without RB injection at the ONH region. After pupil dilation, the optic disc was directly treated with an argon green laser (wavelengths of 532 nm; pore size of 500 μm; power of 80 mW) with 12 pulses and 1 s duration each by a fundus contact lens. The setting of laser treatment was the same as those in our previous paper [36,37].

### 4.11. Statistical Analysis

All the measurements were evaluated in a blinded fashion. Statistical analysis was analyzed by using a commercial software package (GraphPad Prism) (IBM SPSS Statistics 19, International Business Machines Corp., Armonk, NY, USA). The Mann–Whitney U test was used to evaluate differences between groups. Results with *p* values less than 0.05 were considered statistically significant.

## 5. Conclusions

In conclusion, we demonstrated that intravitreal delivery of hWJMSCs in the normal eye resulted in severe retinal inflammation to destroy the retinal architecture, visual function, and retinal function. This intravitreal MSC administration may not be appropriate therapy for retina and optic nerve diseases because the pro-inflammatory cytokines may induce HLA-DR-positive MSCs to trigger rejection in the host. However, the hWJMSC-derived CM provided neuroprotective effects in the experimental model of optic nerve ischemia via anti-inflammation and anti-apoptosis of RGCs. Thus, the MSC-derived secretory factors may be a good candidate for further investigation and development of new approaches for MSC-based therapies in the future.

## Figures and Tables

**Figure 1 ijms-22-02117-f001:**
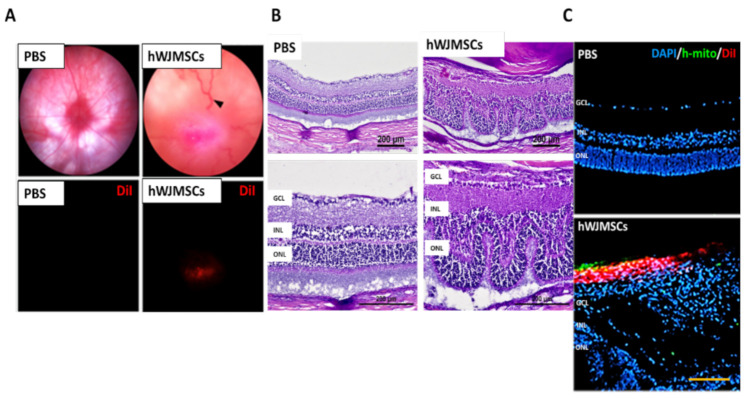
The intravitreal delivery of human Wharton’s jelly mesenchymal stem cells (hWJMSCs) in rat retina four weeks after transplantation. (**A**) Ocular color fundus images in the PBS-treated group and the hWJMSC-treated group. The black arrowhead indicates retinal venous congestion. Red fluorescence shows the DiI-labeled hWJMSCs in the vitreous cavity. Digital magnification (800%) (**B**) Hematoxylin and eosin staining of retinal sections. The hWJMSC transplantation resulted in retinal structural deformity. GCL: ganglion cell layer; INL: inner nuclear layer; ONL: outer nuclear layer. Scale bar 200 µm. (**C**) Immunohistochemistry of human mitochondria staining and DiI-labeled cells in the PBS-treated group and the hWJMSC-treated group. Scale bar 100 µm. DAPI, 4, 6-diamidino-2-phenylindole; h-mito, human-mitochondria; DiI: 1,1’-Dioctadecyl-3,3,3’,3’-Tetramethylindocarbocyanine Perchlorate.

**Figure 2 ijms-22-02117-f002:**
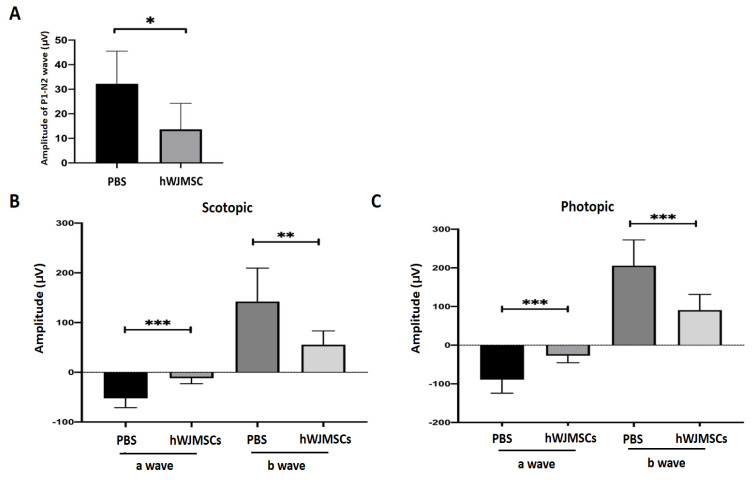
Evaluation of visual function and retinal function through flash visual-evoked potential (FVEP) and electroretinogram (ERG) recordings. (**A**) Bar chart showing the P1-N2 amplitudes in the PBS-treated group and in the hWJMSC group. A significant decrease of P1-N2 amplitude was found in the hWJMSC-treated group as compared to that in the PBS-treated group (*p* < 0.05, *n* = 12). Data are expressed as mean ± SD. P1, P1 the first positive peak; N2, the second negative peak. (**B**) Recordings of scotopic and (**C**) photopic ERG analysis revealed the amplitudes of a- and b-wave in the PBS-treated group and the hWJMSC-treated group. Both visual electrophysiology data showed that the amplitudes of a- and b-waves were significantly reduced by hWJMSC treatment compared to the PBS treatment. Data are expressed as mean ± SD.* *p* < 0.05; ** *p* < 0.01; *** *p* < 0.001.

**Figure 3 ijms-22-02117-f003:**
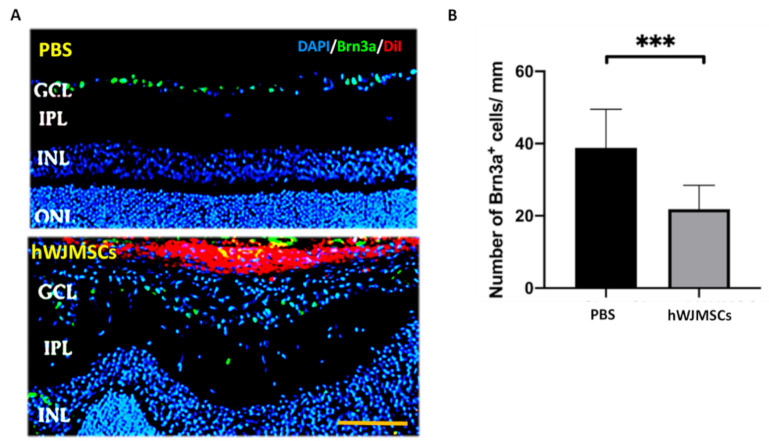
Analysis of the density of Brn3a-positive cell in the PBS-treated group and the hWJMSC-treated group. (**A**) Representative images of Brn3a-positive cells in the GCL. (**B**) Quantification of Brn3a-positive cell density in the PBS-treated group and the hWJMSC group. *** *p* < 0.0001. Scale bar: 100 µm. GCL: ganglion cell layer, IPL: inner plexiform layer, INL: inner nuclear layer; ONL: outer nuclear layer; DAPI, 4, 6-diamidino-2-phenylindole; Brn3a, brain-specific homeobox/POU domain protein 3A; DiI: 1,1’-Dioctadecyl-3,3,3’,3’-Tetramethylindocarbocyanine Perchlorate.

**Figure 4 ijms-22-02117-f004:**
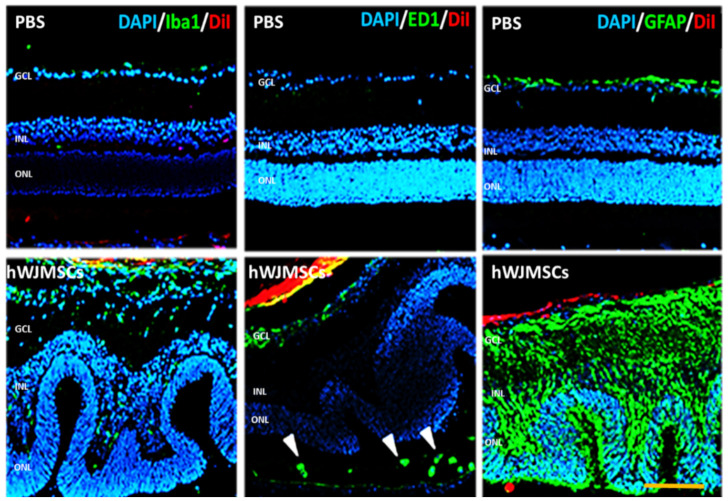
Immunoreactivity of Iba1, ED1, and GFAP in the retina after hWJMSCs transplantation. Representative image of microglia (Iba1+), macrophage (CD68+), and astrocyte (GFAP+) in each group. The confocal image showed the retinal inflammation 28 days after hWJMSCs transplantation. ED1-positive cells infiltrated into the subretinal space (arrowheads). Scale bar: 100 µm. DAPI: 4, 6-diamidino-2-phenylindole; Iba1: ionized calcium binding adaptor molecule 1; ED1: Macrosialin; GFAP: Glial fibrillary acidic protein; DiI: 1,1’-Dioctadecyl-3,3,3’,3’-Tetramethylindocarbocyanine Perchlorate. GCL: ganglion cell layer; INL: inner nuclear layer; ONL: outer nuclear layer.

**Figure 5 ijms-22-02117-f005:**
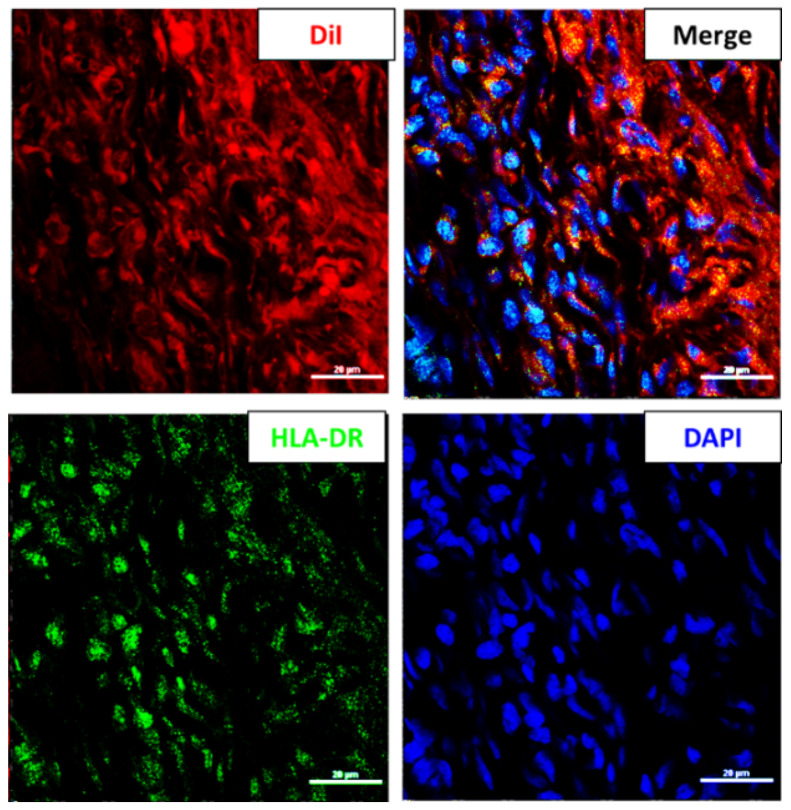
The HLA-DR positive hWJMSC in the vitreous cavity four weeks after transplantation. Representative image of HLA-DR staining in the vitreous cavity. The HLA-DR-positive cells and the nuclei were labeled in green and blue colors, respectively. DiI was used to label the transplanted hWJMSC in red color. Scale bar: 20 µm. DAPI, 4, 6-diamidino-2-phenylindole; DiI: 1,1’-Dioctadecyl-3,3,3’,3’-tetramethylindocarbocyanine perchlorate; HLA-DR: MHC class II antigen.

**Figure 6 ijms-22-02117-f006:**
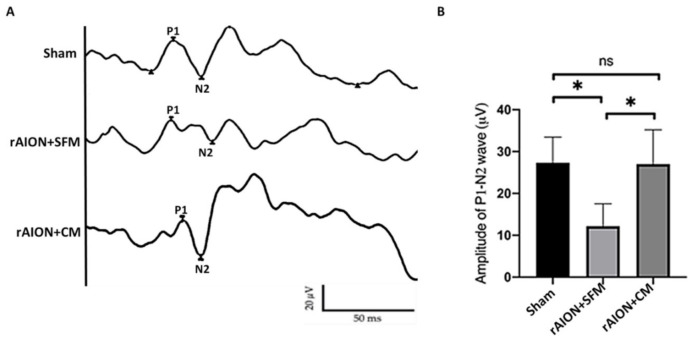
FVEP profiles at day 28 post-rAION (rodent model of anterior ischemic optic neuropathy) induction. (**A**) Representative FVEP wavelet in each group. (**B**) Treatment with CM induced the P1-N2 amplitude by 1.97-fold compared to treatment with SFM (* *p* < 0.05, *n* = 12 per group). Data are expressed as mean ± SD. SFM: serum-free medium; CM: conditioned medium. P1, P1 the first positive peak; N2, the second negative peak; ns, not significant

**Figure 7 ijms-22-02117-f007:**
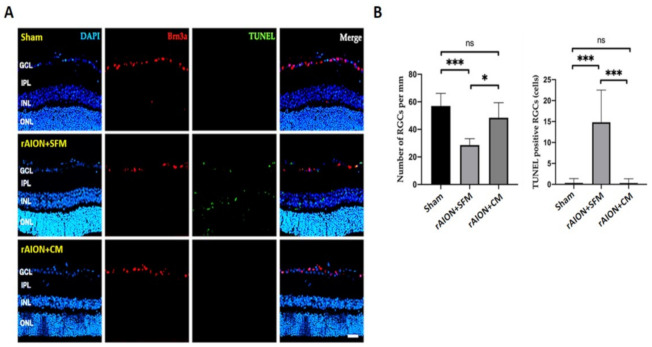
The RGC density and the number of apoptotic RGCs in the 4th week after rAION induction (**A**) Representative images of RGCs (Brn3a-positive cell) in red and TUNEL-positive cells in green. (**B**) Quantification of the RGC density and the number of TUNEL-positive cells. Treatment with CM significantly preserved the RGC density after ON infarct. The apoptotic RGCs were significantly reduced after CM treatment. * *p* < 0.05; *** *p* < 0.001. Scale bar: 40 µm. DAPI, 4, 6-diamidino-2-phenylindole; Brn3a, brain-specific homeobox/POU domain protein 3A; TINEL: Terminal deoxynucleotidyl transferase dUTP nick end labeling; GCL: ganglion cell layer, IPL: inner plexiform layer, INL: inner nuclear layer; ONL: outer nuclear layer.

**Figure 8 ijms-22-02117-f008:**
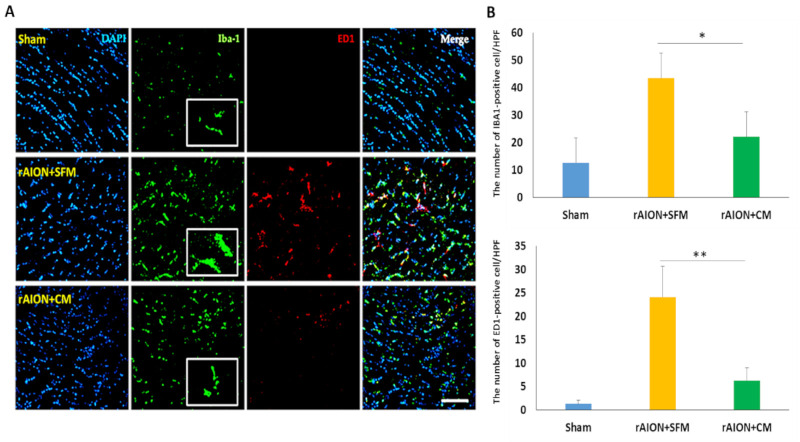
Microglia activation and macrophage infiltration in the optic nerve after ON infarct. (**A**) representative image of Iba1 and ED1 staining in longitudinal sections of the ONs. The Iba1-positive cells, the ED1-positive cells, and the nuclei are marked in green, red, and blue colors, respectively; Treatment with CM effectively reduced the size of the microglia cell (Iba1-positive cell) in contrast to treatment with SFM four weeks after ON infarct. (**B**) Quantification of Iba1+ and ED1+ cells per high-power field (HPF). Data are expressed as mean ± SD (*n* = 6 per group). The decreasing number of Iba1+ cells was significantly different in the CM-treated group as compared to in the SFM-treated group (*n* = 6 per group). The number of macrophages (ED1-positive cells) in the CM-treated group was lower than that in the SFM-treated group (*n* = 6 per group). Scale bar: 100 µm. * *p* < 0.05. ** *p* < 0.01. Iba1: ionized calcium binding adaptor molecule 1; ED1: Macrosialin.

**Figure 9 ijms-22-02117-f009:**
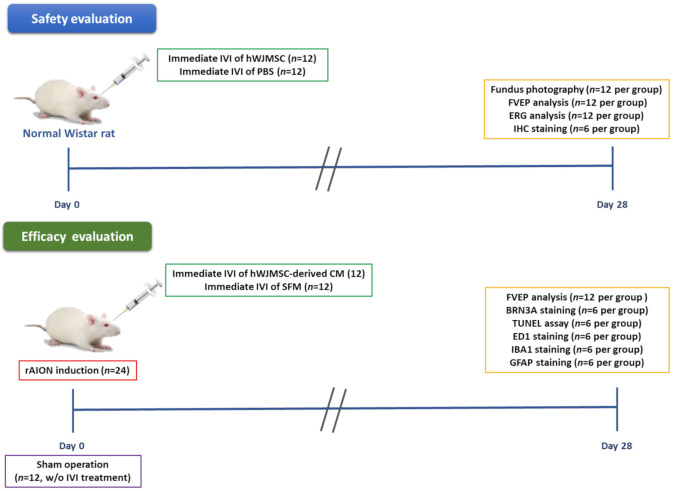
Study design to evaluate the safety and the efficacy of intravitreal MSC-based therapies. The safety of intravitreal injection of hWJMSC was investigated in the normal Wistar rat by using fundus photography, FVEP, ERG, and IHC analysis. The therapeutic effects of intravitreal injection of the hWJMSC-derived CM were evaluated in the rAION model by using FVEP, Brn3a staining, TUNEL assay, ED1 staining, Iba1 staining, and GFAP staining.

## Data Availability

The data presented in this study are available on request from the corresponding author.

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
