# Peer review of "The Benefits and Hazards of Intravitreal Mesenchymal Stem Cell (MSC) Based-Therapies in the Experimental Ischemic Optic Neuropathy"

_ijms, 2021, doi:10.3390/ijms22042117_

Round 1

Reviewer 1 Report

The paper is well-organized and well-written.

The MSC-derived conditioned medium (MDCM) use is certainly preferable to the "allogeneic" MSCs transplantation in the "intravitreal" site.

From the highlighted literature, the problem seems to be caused by the inflammatory reaction induced by allogenicity and inter-species treatment rather than by the MSC cells themselves.

Many studies have shown that MSCs transplantations could yield good results.

However, I acknowledge that the intravitreal administration of both autologous and allogenic MSCs should not be used, as the vitreous is not suitable for this purpose, being a 98.6% gel.

Even more so if the administered MSCs are allogenic and therefore can cause inflammation and reduce their potential therapeutic effects.

In the introduction, the clinical impact of the different implant sites should be described.  

The intravitreal administration, as well as the subretinal and epiretinal ones, are not the route of choice for cell treatments; it should be better specified. Particularly, the intravitreal administration is not a good option, due to the inflammatory risks and the invasiveness of the technique.

The best implantation sites have been shown to be the sub-tenonian and suprachoroidal ones. This is valid for the autologous MSCs. In this way, the MSCs can release their secretome paracrinally without compromising the retinal space.

Preclinical studies, such as this one, should investigate the best way to use cell therapy, the preferable grafting method of MSCs, or how their secretome is administered.

In the clinical practice, intravitreal therapy is always based on drugs, and not on cells. Therefore, the intravitreal MDCM injection could both have a greater clinical benefit and eliminate the safety problems associated with MSCs injection. For this reason, the conclusion of the paper on the MDCM therapeutic perspectives is very interesting. The definition of therapeutic dosages can become useful.

Minor corrections: 

Line 59-64

In ocular diseases, MSC-based therapies provided a potential impact in alternative therapeutics, particularly for chronic diseases, such as retinal degeneration, optic nerve degeneration, glaucoma and uveitis (10). Besides, other therapeutic effects of MSCs proved the potential in the management of ocular surface disease and oculoplastics (10).  Some clinical trials using MSCs are in I-II phase have claimed the safety of intravitreal MSC administration in retinal and optic nerve diseases (10).

A more recent review that corroborates your findings should be cited. For example:

Adak et coll. A Review on Mesenchymal Stem Cells for Treatment of Retinal Diseases. Stem Cell Reviews and Reports. January 2021

Line 240

Although lots of promising and beneficial findings were reported (?) to support MSC therapies, ….

Which reference?

Line 256

However, previous findings demonstrated that MSCs provided vasoprotective and proangiogenic effects in experimental animal models of DR and other retinal diseases (29-30).

There are no citations 29 and 30 among the references and the acronym of DR is missing.

Line 454

Abbreviations can be implemented.

Line 517-530

  1. Huang TL, Wen YT, Ho YC, Wang JK, Lin KH, Tsai RK. Algae Oil Treatment Protects Retinal Ganglion Cells (RGCs) via ERK Signaling Pathway in Experimental Optic Nerve Ischemia. Mar Drugs. 2020 Jan 27;18(2):83. doi: 10.3390/md18020083. PMID: 32012745; PMCID: PMC7074556.
  2. Liu, PK., Wen, YT., Lin, W. et al. Neuroprotective effects of low-dose G-CSF plus meloxicam in a rat model of anterior ischemic optic neuropathy. Sci Rep 10, 10351 (2020). https://doi.org/10.1038/s41598-020-66977-9

References 26 and 27 are not mentioned in the text.

Author Response

The paper is well-organized and well-written. The MSC-derived conditioned medium (MDCM) use is certainly preferable to the "allogeneic" MSCs transplantation in the "intravitreal" site. From the highlighted literature, the problem seems to be caused by the inflammatory reaction induced by allogenicity and inter-species treatment rather than by the MSC cells themselves. Many studies have shown that MSCs transplantations could yield good results. However, I acknowledge that the intravitreal administration of both autologous and allogenic MSCs should not be used, as the vitreous is not suitable for this purpose, being a 98.6% gel. Even more so if the administered MSCs are allogenic and therefore can cause inflammation and reduce their potential therapeutic effects.

Reply:

We appreciated that the reviewer 1 agreed with our opinion and gave us many useful suggestions to improve this manuscript.

In the introduction, the clinical impact of the different implant sites should be described. The intravitreal administration, as well as the subretinal and epiretinal ones, are not the route of choice for cell treatments; it should be better specified. Particularly, the intravitreal administration is not a good option, due to the inflammatory risks and the invasiveness of the technique. The best implantation sites have been shown to be the sub-tenonian and suprachoroidal ones. This is valid for the autologous MSCs. In this way, the MSCs can release their secretome paracrinally without compromising the retinal space.

Reply:

We have added the clinical impact of different implant sites in the introduction section as following:

Mechanical stress during the transplantation procedure or lack of an optimized dosage and protocol for transplantation may result in cell death at transplanted sites and potential risks. Recent clinical trials reported that suprachoroidal or sub-tenon MSCs implantation are relative safe routes for retinal degeneration treatment [25, 26, 27]. In another clinical trial, the outcomes of subretinal MSCs implantation in 11 patients were reported that there were no systemic complications but 6 patients experienced ocular complications [28]. Till now, the epiretinal MSCs transplantation is not applied in clinic. Notably, the safety of intravitreal MSCs implantation is still contraversail. A prospective, phase I, clinical trial reported that intravitreal injection of autologous bone marrow MSCs into patients' eye does not meet safety standards. Major side effects of this therapy can include fibrosis and TRD [29]. However, Aekkachai Tuekprakhon et al., reported that intravitreal injection of BM-MSCs appears to be safe and potentially effective. All adverse events during the 12-month period required observation without any intervention. For the long-term follow-up, only one participant needed surgical treatment for a serious adverse event and the vision was restored [30]. Taken together, the proper MSCs implantation routes in eye are considered be the sub-tenonian and suprachoroidal ones. The safety of subretinal, epiretinal, and intravitreal implantations of MSCs may need further evaluation before medical treatment to patients.

Preclinical studies, such as this one, should investigate the best way to use cell therapy, the preferable grafting method of MSCs, or how their secretome is administered. In the clinical practice, intravitreal therapy is always based on drugs, and not on cells. Therefore, the intravitreal MDCM injection could both have a greater clinical benefit and eliminate the safety problems associated with MSCs injection. For this reason, the conclusion of the paper on the MDCM therapeutic perspectives is very interesting. The definition of therapeutic dosages can become useful.

Reply:

We agreed with the reviewer 1’s comment. In our study, the intravitreal injection with MDCM provided a beneficial outcome and eliminated the adverse effects in rAION. Based on these findings, we are interested in investigating the dosage and component of MDCM in our future works. Thus, we have designed the secretome and exosomal miRNA analyses in our recent studies. We expected the omic analyses can reveal the key factor in the CM for optimizing the dosage of key factor in the rAION model.  

Minor corrections:

Line 59-64

In ocular diseases, MSC-based therapies provided a potential impact in alternative therapeutics, particularly for chronic diseases, such as retinal degeneration, optic nerve degeneration, glaucoma and uveitis (10). Besides, other therapeutic effects of MSCs proved the potential in the management of ocular surface disease and oculoplastics (10). Some clinical trials using MSCs are in I-II phase have claimed the safety of intravitreal MSC administration in retinal and optic nerve diseases (10). A more recent review that corroborates your findings should be cited. For example: Adak et coll. A Review on Mesenchymal Stem Cells for Treatment of Retinal Diseases. Stem Cell Reviews and Reports. January 2021

Reply:

Thanks for your suggestions. We have added the review article in the revised manuscript as following:

Besides, other therapeutic effects of MSCs proved the potential in the management of ocular surface disease and oculoplastics [10, 11]. Some clinical trials using MSCs are in I/II phase have claimed the safety of intravitreal MSC administration in retinal and optic nerve diseases [10, 11].

However, there is still a long way to go to prove the efficacy and therapeutic effects in patients. Especially, lack of integration of MSCs and induction of reactive gliosis after intravitreal injection were also reported in some studies [11, 15-17].

Line 240

Although lots of promising and beneficial findings were reported (?) to support MSC therapies, …. Which reference?

Reply:

Thanks for your correction. We have added the reference in the sentence as following:

Although lots of promising and beneficial findings were reported to support MSC therapies [10-14], some adverse effects were demonstrated in recent studies, including detrimental outcomes in stem cell therapy against eye diseases [15-17].

Line 256

However, previous findings demonstrated that MSCs provided vasoprotective and proangiogenic effects in experimental animal models of DR and other retinal diseases (29-30). There are no citations 29 and 30 among the references and the acronym of DR is missing.

Reply:

Thanks for your correction. We have revised this sentence in the discussion section as following:

However, previous findings demonstrated that MSCs provided vasoprotective and proangiogenic effects in experimental animal models of diabetic retinopathy and other retinal diseases [33, 34].

Line 454

Abbreviations can be implemented.

Reply:

We have corrected this reference format as following:

Millán-Rivero, J. E.; Nadal-Nicolás, F. M.; García-Bernal, D.; Sobrado-Calvo, P.; Blanquer, M.; Moraleda, J. M.; Vidal-Sanz, M.; Agudo-Barriuso, M., Human Wharton's jelly mesenchymal stem cells protect axotomized rat retinal ganglion cells via secretion of anti-inflammatory and neurotrophic factors. Scientific Report 2018, 8, (1), 16299.

Line 517-530

Huang TL, Wen YT, Ho YC, Wang JK, Lin KH, Tsai RK. Algae Oil Treatment Protects Retinal Ganglion Cells (RGCs) via ERK Signaling Pathway in Experimental Optic Nerve Ischemia. Mar Drugs. 2020 Jan 27;18(2):83. doi: 10.3390/md18020083. PMID: 32012745; PMCID: PMC7074556.

Liu, PK., Wen, YT., Lin, W. et al. Neuroprotective effects of low-dose G-CSF plus meloxicam in a rat model of anterior ischemic optic neuropathy. Sci Rep 10, 10351 (2020). https://doi.org/10.1038/s41598-020-66977-9

References 26 and 27 are not mentioned in the text.

Reply:

Thanks for your correction. We have corrected these two citations in the revised manuscript. These two references are cited in the method section as following:

Eyeballs were enucleated, surrounding connective tissues and muscles were removed, the cornea was cut with micro scissors to improve the fixation process [38, 39].

The procedure of immunohistochemistry was described in detail in our previous reports [38, 39].

The detailed procedure of rAION induction was described in our previous report [38, 39].

The setting of laser treatment was the same as those in our previous paper [38, 39].

Reviewer 2 Report

The manuscript “The benefits and hazards of intravitreal mesenchymal stem cell (MSC) based-therapies in the experimental ischemic optic neuropathy” by Yao-Tseng Wen and co-authors evaluates potential risks of an intravitreal mesenchymal stem cell application using an animal model. The manuscript is well written, but some improvements, especially to the figures, could be made.

Please introduce the abbreviation MSC (abstract) before using it.

Please include the animal care committee approval number. Also include ethical regulation information about the human umbilical cord cells used, including a ethical committee approval number.

Authors need to include more information about immunohistochemistry. How many section per retina were stained e.g. with Brn-3a? in which area of the sections were photos taken and how were cell counts performed. Did they also evaluate RGC counts using RT-qPCR or Western Blot? Also, how were the TUNEL+ cells quantified. Microglia cells should also be quantified.

Add precise p-values to the results section, not just p<0.05.

Some of the labelling in the figures is hard to read read, like “DiI” in figure 1A. Also, I can not make out the writing above the scale bar in figure 1B. The photo labelling is quite inconsistent, the authors use different colors for scale bars, sometimes groups are written next to the photos and sometimes in the photos, in some cases the retina layer description is missing (figure 1C, figure 4). this needs to be carefully revised and needs to be more considered. The appear stretched.  

Something is wrong with the graphs in figure 2b and C.  

Figure 5: add the size of the scale bar to the legend.

Include graphs of microglia cell counts.

Authors should further discuss that MSCs come from a different species and if this could affect study results.

Minor comments:

-line 37: remove extra space after sentence.

- BRN3A – correct to Brn3a for staining (protein name)

Author Response

The manuscript “The benefits and hazards of intravitreal mesenchymal stem cell (MSC) based-therapies in the experimental ischemic optic neuropathy” by Yao-Tseng Wen and co-authors evaluates potential risks of an intravitreal mesenchymal stem cell application using an animal model. The manuscript is well written, but some improvements, especially to the figures, could be made.

Reply:

We appreciated that the reviewer 2 gave us this positive comment. The point by point responses were addressed below.

Please introduce the abbreviation MSC (abstract) before using it.

Reply:

Thanks for your correction of the abbreviation in the abstract. We have modified the sentence in the revised manuscript as following:

Mesenchymal stem cell (MSC) therapy was investigated intensively for many years. However, there is a potential risk related to MSC applications in various cell niches.

Please include the animal care committee approval number. Also include ethical regulation information about the human umbilical cord cells used, including an ethical committee approval number.

Reply:

Thanks for your constructive suggestion. We have added the animal care committee approval number and the ethical committee approval number in the revised manuscript as following:

Besides, all animal experiments were approved by the Institutional Animal Care and Use Committee of Buddhist Tzu Chi General Hospital (106-21-2).

The hWJMSCs used in this study were approved by the Research Ethics Committee of Buddhist Tzu Chi General Hospital, and written informed consent was obtained from all participants (Institutional Review Board 100-166).

Besides, the regulations about human sample were described in the method section as following:

All experiments related to the hWJMSCs were followed by the institutional guidelines and approved by the Research Ethics Committee, Hualien Tzu Chi Hospital.

Authors need to include more information about immunohistochemistry. How many section per retina were stained e.g. with Brn-3a? in which area of the sections were photos taken and how were cell counts performed. Did they also evaluate RGC counts using RT-qPCR or Western Blot? Also, how were the TUNEL+ cells quantified. Microglia cells should also be quantified.

Reply:

We have added more information of immunohistochemistry in the method section. In addition, we didn’t use the RT-qPCR and Western blotting analyses to count RGC density in this study. The number of microglial cell was reported in the Figure 8 and the result section.

Add precise p-values to the results section, not just p<0.05.

Reply:

We have added the p value in the revised manuscript as following:

The amplitude of the P1–N2 in the PBS-treated group was higher than that in the hWJMSC-treated group (p = 0.0127).

The scotoptic ERG recordings demonstrated that a-wave and b-wave amplitudes were significantly decreased in the hWJMSC-treated group compared to those in the PBS-treated group (Figure 2B; a-wave, p = 0.0002; b-wave, p = 0.0062 ). In the photopic ERG analysis, a-wave and b-wave amplitudes were also significantly decreased in the hWJMSC-treated group compared to those in the PBS-treated group (Figure 2C; a-wave, p = 0.0002; b-wave, p = 0.0003).

Some of the labelling in the figures is hard to read, like “DiI” in figure 1A. Also, I can not make out the writing above the scale bar in figure 1B. The photo labelling is quite inconsistent, the authors use different colors for scale bars, sometimes groups are written next to the photos and sometimes in the photos, in some cases the retina layer description is missing (figure 1C, figure 4). this needs to be carefully revised and needs to be more considered. The appear stretched. 

Reply:

We appreciated your corrections to improve the quality of this study. We have modified that figure 1 in the revised manuscript. We also make the labels to be consistent in all figures.

Something is wrong with the graphs in figure 2b and C. 

Reply:

Thanks for your correction. We have modified the figure legend in the revised manuscript as following:

(B) Recordings of scotopic and (C) photopic ERG analysis revealed the amplitudes of a wave and b wave in the PBS-treated group and the hWJMSC-treated group.

Figure 5: add the size of the scale bar to the legend.

Reply:

We have added the size of scale bar in the legend.

Include graphs of microglia cell counts.

Reply:

We have added the graphs of macrophage cell counts and microglia cell counts in the Figure 8B.

Authors should further discuss that MSCs come from a different species and if this could affect study results.

Reply:

Thanks for your suggestion. We have discussed the inter-species issue in the discussion section as following:

However, HLA-DR expression in the allogeneic MSCs may induce allogeneic rejection in the host. Especially, interspecies MSCs transplantation might cause much severe inflammatory response because the diseased ocular environment in these animal models might not be identical to the human environment. We considered that the autologous MSCs implantation in the vitreous cavity may prevent this severe inflammatory response. Thus, intravitreal injection of autologous MSCs needs futher investigation in a safe and healthy manner.

Minor comments:

line 37: remove extra space after sentence.

Reply:

Thanks for your kind correction. We have removed the space in the revised manuscript.

In middle-aged person, the most common type of acute optic neuropathy is non-arteritic anterior ischemic optic neuropathy (NAION), with an incidence of 3.72 per 100 000 in Taiwan and at least 6000 new cases a year [1]. Transient nonperfusion or hypoperfusion of the optic nerve (ON) disc head (ONH) is the most common pathogenesis of NAION [2].

BRN3A – correct to Brn3a for staining (protein name)

Reply:

Thanks for your correction. We have changed the name as your suggestion.

Round 2

Reviewer 2 Report

The authors made several improvements to their manuscript. There are a few minor issues left.

Figure 2 B and C need to be revised again, they still look pulled apart.

The authors still write BRN3A-positive cells, this needs to be Brn3a-positve cells and IBA1 needs to be corrected to Iba1.

Figure 3. A - also write Brn3a on the photo, not Brn.

Author Response

The authors made several improvements to their manuscript. There are a few minor issues left.

Reply:

Thanks for your careful revision. We have made the corrections as your suggestion in the revised manuscript.

Figure 2 B and C need to be revised again, they still look pulled apart.

Reply:

The Figure 2 Band C have been revised in the result section to reduce stretched image.

The authors still write BRN3A-positive cells, this needs to be Brn3a-positve cells and IBA1 needs to be corrected to Iba1.

Reply:

We have corrected the BRN3A and IBA1 in the revised manuscript.

Figure 3. A - also write Brn3a on the photo, not Brn.

Reply:

Thanks for your correction. We have made this correction in the Figure 3A.

.
